# Development of LPFG-Based Seawater Concentration Monitoring Sensors Packaged by BFRP

**DOI:** 10.3390/mi16070810

**Published:** 2025-07-14

**Authors:** Zhe Zhang, Tongchun Qin, Yuping Bao, Jianping He

**Affiliations:** School of Civil Engineering, Nantong Institute of Technology, Nantong 226000, China; zhangzhe_nt@126.com (Z.Z.); qintc_nt@sina.com (T.Q.); baoyp_nt@126.com (Y.B.)

**Keywords:** LPFG, seawater concentration, calcium chloride solution concentration, sodium chloride solution concentration, structural health monitoring

## Abstract

Leveraging the sensitivity of long-period fiber grating (LPFG) to changes in the environmental refractive index, an LPFG-based seawater concentration monitoring sensor is proposed. Considering the highly saltine and alkali characteristics of the sensor’s operating environment, the proposed sensor is packaged by basalt fiber-reinforced polymer (BFRP), and the sensor’s sensitivities were studied by sodium chloride and calcium chloride solution concentration experiments and one real-time sodium chloride solution concentration monitoring experiment. The test results show the wavelength of LPFG, a 3 dB bandwidth and a peak loss of LPFG’s spectrogram change with changes in the concentration of sodium chloride or calcium chloride solutions, but only the wavelength has a good linear relationship with the change in solution concentration, and the sensing coefficient is −0.160 nm/% in the sodium chloride solution and −0.225 nm/% in the calcium chloride solution. The real-time monitoring test further verified the sensor’s sensing performance, with an absolute measurement error of less than 1.8%. The BFRP packaged sensor has good corrosion resistance and a simple structure, and it has a certain application value in the monitoring of salinity in the marine environment and coastal soil.

## 1. Introduction

Spanning approximately 360 million square kilometers, ocean covers 71% of the planet’s surface and contains 97% of its water. As a dominant force in global weather and climate systems, even minor oceanic temperature fluctuations can trigger significant climatic shifts worldwide. Salinity variations further influence marine ecosystems, altering species distribution and estuarine ecology—high-salinity zones favor biologically adapted organisms, while low-salinity regions support greater biodiversity. Moreover, salinity changes impact the global water cycle, exacerbating droughts and precipitation extremes. Given these critical roles, salinity stands as a key parameter for seawater characterization and marine environmental monitoring.

Seawater salinity monitoring currently employs the following primary techniques: the interference method [1], conductivity measurements [2,3], microwave remote sensing [4,5], and optical fiber sensing [6,7,8]. Among these methods, the interference method is one of the most sensitive methods for measuring the refractive index of seawater, which has a wide-ranging adaptability (refractive index measurement range of 1.33–1.38); the remote sensing salinity monitoring systems are usually based on single-band algorithms or dual-band dual-polarization algorithms to invert salinity, with the advantage of a large measurement range and continuous observation, and the disadvantage of low measurement accuracy and data inversion susceptible to interference; the electrode-based conductivity measurement technology has the advantages of a high measurement accuracy and stable performance, with the disadvantage of a shorter service life. Furthermore, Esiyu et al. developed a two-channel surface plasmon resonance sensor for simultaneous measurement of seawater salinity and temperature, and the salinity sensitivity and temperature sensitivity of the sensor could reach 0.3769 nm/ ^0^/_00_ and −0.956 nm/°C, respectively [9]. Optical fiber sensors have the characteristics of a high sensitivity, good stability and resistance to humidity, and have been successfully applied for structural safety monitoring [10,11,12]. In the salinity monitoring field, Sun et al. developed a fiber Bragg grating (FBG) salinity sensor coated with polyimide (PI) for in situ monitoring of groundwater salinity [13]. Luo et al. developed an EFBG (etched FBG) to measure salinity and temperature simultaneously [14]. FBG is used for salinity measurement, mainly by encapsulating or coating salinity-sensitive materials in its indication, which has a limitation of having a unidirectional sensitivity (only detects increasing salinity trends). Seawater fiber-optic salinity sensors based on the refractive index detection principle can effectively monitor seawater salinity [15,16]. Among these, LPFG is particularly sensitive to the ambient refractive index and has been used for monitoring corrosion, solution concentration, etc. For example, Tang et al. developed an LPFG-FBG sensor installed on steel rebars for simultaneous measurement of corrosion and strain [17]. Chen et al. and Tang et al. developed two types of LPFG corrosion sensors coated with Ag and Fe-C layers, respectively, which can monitor corrosion loss up to different ranges [18,19]. In these studies, the LPFG surface was modified with silver (Ag) and iron-carbon (Fe-C) coating layers. As these protective layers undergo corrosion, they induce measurable changes in the surrounding refractive index at the LPFG surface. The corrosion loss is then quantitatively determined through a pre-established correlation between the center wavelength shift and the corrosion coefficient. The disadvantage of this sensor is that it cannot be reused. The disadvantage of this sensor is that it cannot be reused. Ambient temperature variations induce shifts in the LPFG’s center wavelength, potentially interfering with other parameter measurements, and as the ambient temperature changes, thermo-optic effects cause corresponding shifts in the LPFG’s center wavelength. This temperature sensitivity must be accounted for to prevent cross-talk in multi-parameter sensing applications. In current sensing and monitoring applications, dual-parameter compensation has emerged as the predominant approach for temperature compensation. This methodology typically employs two distinct sensing mechanisms: one dedicated to temperature measurement and another targeting the specific parameter of interest. Through differential signal processing, the temperature-induced variations can be effectively decoupled from the target parameter measurements, thereby achieving accurate temperature compensation [20]. To remove the effect of ambient temperature, Qi et al. designed a novel cascaded grating sensor with film-modified LPFG and FBG. In this study, The LPFG exhibits dual sensitivity to both temperature and humidity variations, whereas the FBG responds solely to temperature changes. By utilizing the FBG as a dedicated temperature reference, the coupled temperature–humidity response of the LPFG can be decoupled through differential signal processing, thereby enabling effective temperature compensation for humidity measurements [21]. Marine environments contain high concentrations of ionic compounds (e.g., NaCl, Mg^2+^, Ca^2+^, K^+^) and trace elements that accelerate electrochemical corrosion. Conventional metal enclosures using stainless steel or aluminum alloys demonstrate limited durability in these conditions, exhibiting progressive material degradation that compromises sensor longevity and measurement accuracy. In contrast, fiber-reinforced polymer composites demonstrate exceptional chemical stability, maintaining structural integrity in acidic, alkaline, and saline environments, which makes them ideal for long-term deployment in aggressive marine applications. Among fiber-reinforced polymers, each material exhibits distinct characteristics: GFRP (Glass Fiber-Reinforced Polymer) is economical but suffers from brittleness and limited thermal stability; CFRP (Carbon Fiber-Reinforced Polymer) offers excellent high-temperature resistance and electrical conductivity, though at a higher cost and with similar brittleness concerns; and BFRP (Basalt Fiber-Reinforced Polymer) combines superior mechanical strength, exceptional corrosion resistance, and thermal durability, making it particularly suitable for civil and marine engineering applications [22,23]. Recent studies have demonstrated BFRP’s effectiveness in optical sensor encapsulation. For example, Wu et al. investigated BFRP-encapsulated FBG sensors, confirming their fatigue resistance and durability under railway loading conditions [24]. Liu et al. characterized the mechanical and sensing performance of OFBG-BFRP composites in alkaline concrete environments [25].

To address the critical need for accurate seawater monitoring, this study develops a novel salinity sensor based on long-period fiber grating (LPFG) technology. The sensor employs basalt fiber-reinforced polymer (BFRP) encapsulation to enhance both durability and measurement stability in marine environments. The experimental validation comprises three systematic investigations: (1) controlled sodium chloride (NaCl) concentration tests, (2) calcium chloride (CaCl_2_) solution measurements, and (3) real-time dynamic monitoring of NaCl concentration variations. This comprehensive approach quantitatively evaluates the sensor’s sensitivity and linear response characteristics for practical seawater monitoring applications.

## 2. Basic Theory of Long Period Fiber Gratings

The theory of LPFG is developed based on the theory of fiber Bragg grating (FBG), which is a kind of fiber grating with a period of hundreds of microns, featuring low insertion loss, small backscattering, simple fabrication as well as a low cost. It has a good application prospect in the field of fiber optic sensing and optical communication. The mode coupling of long-period fiber grating mainly refers to the coupling between the core fundamental mode LP_01_ and the cladding modes LP_0m_ (m = 2, 3, 4, …) of each order transmitted in the same direction. From the theory of coupled modes, the phase matching condition of the long-period fiber grating can be expressed as follows:(1)$$\lambda_{D}^{m}=(n_{eff}^{co}-n_{eff}^{cl,m})\Lambda$$

Here, $\lambda_{D}^{m}$ is resonant wavelength; Λ is LPFG’s period; $n_{eff}^{co},n_{eff}^{cl,m}$ are effective refractive indexes of the core basemodule (LP_01_) and the first-order m-subcladding mode (LP_0m_), respectively [26,27].

Figure 1 is the long-period fiber grating three-layer step fiber model, a,b are the core and cladding radius, respectively; $n_{1},n_{2},n_{3}$ are the core, cladding and cladding outside the refractive index of the medium, respectively. The resonance wavelength and transmitted wave-loss peak amplitude of the LPFG are very sensitive to changes in the external environment, with higher temperature, bending, twisting, and transverse load sensitivities than FBG. It has been shown that the resonant wavelength of the LPFG is related not only to the period of the grating and the refractive index of the core, but also to the effective refractive index of the cladding mode, which is related to the radius of the cladding mode and the refractive index of the outer medium (environment).

## 3. Sensor Design

Figure 2a shows the structural schematic diagram of the LPFG-based seawater concentration monitoring sensor, which includes one LPFG, sponge, BFRP (Basalt Fiber Reinforced Polymer) shell and optical fiber. LPFG is used to measure the seawater concentration, the sponge is used to isolate marine microorganisms and reduce interference with LPFGs from external stresses, and BFRP is used as a housing for sensors, mainly because BFRP is less susceptible to corrosion in marine environments, thus increasing the service life of the sensor. Figure 2b is a photograph of the LPFG-based seawater concentration-monitoring sensor, of which the length is 120 mm and the diameter is 5 mm. In the sensor, the LPFG is a long-period fiber grating written by high-frequency CO_2_ laser pulses, and LPFG is pre-stretched and fixed in a BFRP shell which can eliminate the effects of strain, bending and twisting on measurement results. Here, the BFRP shell was fabricated by 3D print technology. BFRP has excellent corrosion resistance and is not easily corroded by chemical media such as acids, alkalis, and salts. It has a tensile strength of 722 MPa, ensuring that it can be used for a long time in environments with acids, alkalis, salts, and certain pressures.

## 4. Sodium Chloride and Calcium Chloride Concentration Measuring Test

Figure 3 depicts the schematic diagram of the test setup including one FBG demodulator (SI720 produced by MOI), one thermostat box and one LPFG-based seawater concentration-monitoring sensor. Figure 4 shows the temperature-sensing characteristic of the LPFG-based seawater concentration-monitoring sensor. The temperature sensing coefficients are about 0.6205 nm/°C and 0.06461 nm/°C, respectively.

Sodium chloride and calcium chloride are the main components of seawater, and the concentration of seawater can be broadly reflected by monitoring the concentration of sodium chloride or calcium chloride solutions. In the thermostat box, some sodium chloride solution and calcium chloride solution are, respectively, stored in the thermostat box and an LPFG-based seawater concentration monitoring sensor is placed into the sodium chloride solution and calcium chloride solution.

### 4.1. Sodium Chloride Solution Concentration Measurement Test

The initial center wavelength (resonant wavelength) of the long period in air is 1564 nm with a period of 600 um, a peak loss of −13.609 dB, and a 3 dB bandwidth of 4.976 nm. The temperature in the thermostat box is always 20 °C. The solubility of NaCl at 20 °C is 36 g. NaCl does not react with water during dissolution and there is no heat exchange. Throughout the test, sodium chloride was gradually added to water to form a sodium chloride solution of a certain concentration.

To obtain the linear relationship between the NaCl solution concentration and the central wavelength of LPFG, two tests were conducted. The maximum concentration of the NaCl solution is about 17%. Figure 5 shows the test results and the central wavelength of the LPFG; the NaCl solution concentration has a good linear relationship; the linear sensitivity coefficients of the three tests are −0.157 nm/%, −0.160 nm/% and −0.163, respectively; the linear correlation coefficients are −0.998, −0.994 and −0.999, respectively. The negative sensitivity coefficient indicates that the central wavelength of LPFG shifts to the short-wave direction with the increase in the NaCl solution concentration.

The relationship between the NaCl solution concentration and the spectral characteristics of the LPFG is illustrated in Figure 6 and Figure 7. Figure 6 demonstrates the variation in the 3 dB bandwidth of the LPFG’s spectrogram with respect to NaCl concentration, while Figure 7 presents the corresponding trend for the peak loss. As observed, the peak loss exhibits a discernible increase with higher NaCl concentrations, though the correlation lacks strong linearity. The 3 dB bandwidth displays irregular fluctuations in response to changes in NaCl concentration.

### 4.2. Calcium Chloride Solution Concentration Measurement Test

After thoroughly rinsing the LPFG-based sensor and the thermostat box with deionized water, the thermostat box was filled with clean water, and the sensor was immersed in the solution. Calcium chloride (CaCl_2_) particles were gradually introduced to vary the solution concentration, while the SI720 optical interrogator recorded the corresponding wavelength shifts at each concentration step. The dissolution of calcium chloride (CaCl_2_) particles in water is an exothermic process, leading to a transient temperature increase in the solution. To maintain measurement accuracy, the solution was allowed to thermally equilibrate to the baseline temperature prior to each spectral measurement. Figure 8 shows the test results. It can be seen that the LPFG can sense the change in CaCl_2_ concentration well, the center wavelength of the LPFG and the CaCl_2_ solution concentration also has a good linear relationship, and the linear sensitivity coefficients of the three tests were −0.230 nm/%, −0.222 nm/% and −0.223 nm/%, respectively; the linear correlation coefficients are −0.998, −0.993 and −0.999, respectively, as listed in Table 1.

Figure 9 and Figure 10 depict the influence of CaCl_2_ solution concentration on the spectral response of the LPFG. Specifically, Figure 9 illustrates the variation in the 3 dB bandwidth of the LPFG’s spectrogram, while Figure 10 presents the corresponding trend for the peak loss. Similar to the observations for NaCl, the peak loss increases with higher CaCl_2_ concentrations, though the relationship exhibits limited linearity. In contrast, the 3 dB bandwidth undergoes irregular shifts as the CaCl_2_ concentration changes, indicating a non-monotonic dependence. The results of the above calibration tests show that although the center wavelength, spectral 3 dB bandwidth, and spectral peak loss of the sensor are all related to changes in solution concentration, only the center wavelength is linearly related to solution concentration.

### 4.3. Real-Time Sodium Chloride Solution Concentration Monitoring Test

In this test, one real-time monitoring test for sodium chloride solution concentration was conducted. Specifically, the dissolution process of sodium chloride (NaCl) was monitored in real time using the correlation between the NaCl solution concentration and the central wavelength shift of the long-period fiber grating (LPFG). The experiment was conducted at a constant ambient temperature of 26 °C, with 150 g of NaCl dissolved in 500 mL of deionized water, which was left to stand indoors for one hour beforehand to ensure that its temperature was consistent with the room temperature, yielding a final solution concentration of 23.1%. The real-time monitoring results are presented in Figure 11. After one hour of continuous measurement, the LPFG system recorded a NaCl solution concentration of 24.9%, representing a 1.8% absolute error compared to the expected value. This measurement discrepancy primarily arises from two factors: (1) slight variations in ambient temperature and (2) inherent measurement errors in the LPFG system. A particularly notable observation was an initial rapid concentration spike from 0% to approximately 27% immediately following NaCl introduction. This transient phenomenon can be attributed to two concurrent effects: first, the mechanical disturbance caused by salt particle addition induced water turbulence, temporarily altering the stress distribution on the LPFG sensor; second, the presence of undissolved micro-particles modified the solution’s local refractive index. The LPFG’s characteristic sensitivity to refractive index variations explains this observed signal perturbation during the initial dissolution phase. The experimental data clearly demonstrate a consistent blueshift in the LPFG’s central wavelength with increasing NaCl concentration, in excellent agreement with prior calibration results. Utilizing the established linear correlation between NaCl concentration and wavelength shift (Figure 11b), we can precisely determine solution concentration through real-time LPFG measurements. Figure 12 shows the real-time monitoring data for solution temperature measured by LPFG sensor encapsulated in a capillary tube, and the sampling time is every 300 s. It can be seen that the solution temperature is basically maintained at 26 °C.

## 5. Conclusions

In this study, we employ basalt fiber-reinforced polymer (BFRP) to encapsulate long-period fiber grating (LPFG) for developing a seawater-concentration monitoring sensor. Through systematic experiments, we investigate the sensor’s sensitivity and real-time measurement performance, yielding the following key findings:(1)The temperature coefficient of the sensor in freshwater is about 0.0633 nm/°C;(2)The LPFG exhibits sensitivity to NaCl and CaCl_2_ solution concentration changes through three parameters: center wavelength, peak loss at center wavelength, and 3 dB bandwidth. However, only the center wavelength shows a strong linear correlation with concentration variation. The linear coefficients are determined to be −0.160 nm/% for NaCl solution and −0.225 nm/% for CaCl_2_ solution, respectively.(3)The developed sensor achieves real-time monitoring of NaCl solution concentration changes with a measurement error of approximately 1.8%, demonstrating good practical applicability.

Previous studies have shown that the BFRP encapsulation demonstrates excellent corrosion resistance in seawater environments, ensuring long-term stability for sensor operation. This study does not account for potential interference from environmental factors such as vibration (e.g., wave impacts) or temperature variations on the sensor’s concentration measurements. Future research will focus on refining the sensor design and conducting multi-parameter coupling tests to analyze these effects comprehensively.

## Figures and Tables

**Figure 1 micromachines-16-00810-f001:**
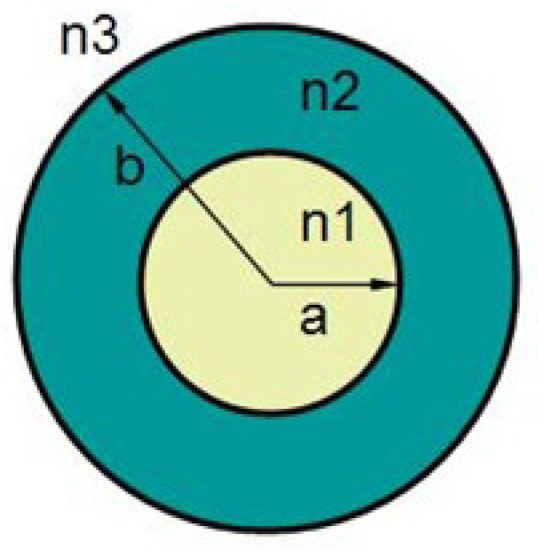
Cross-section of LPFG.

**Figure 2 micromachines-16-00810-f002:**
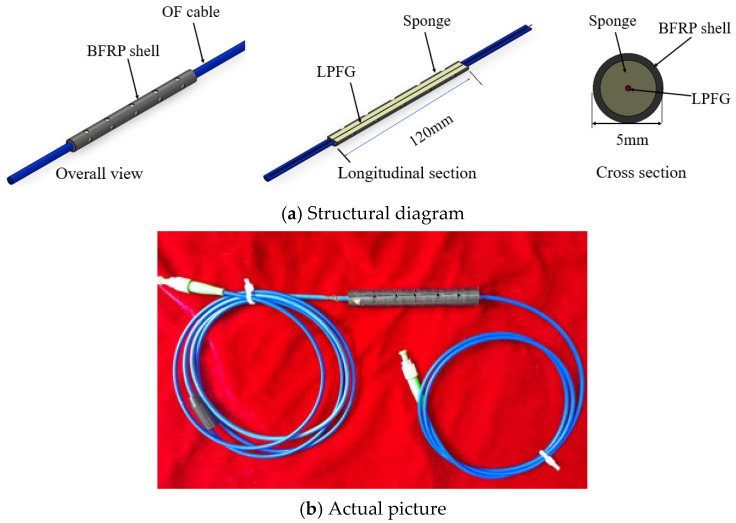
The structural schematic and actual diagrams of the LPFG-based seawater concentration monitoring sensor.

**Figure 3 micromachines-16-00810-f003:**
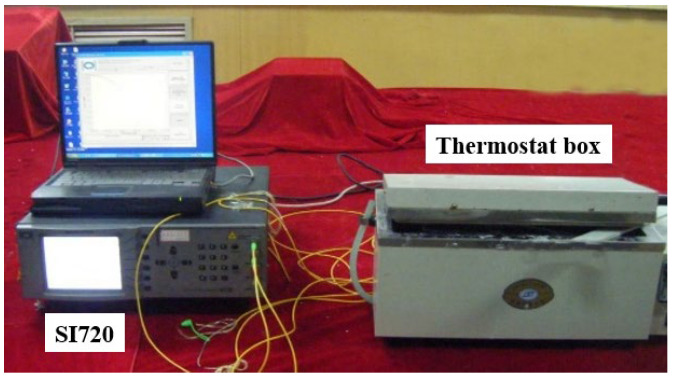
Test setup.

**Figure 4 micromachines-16-00810-f004:**
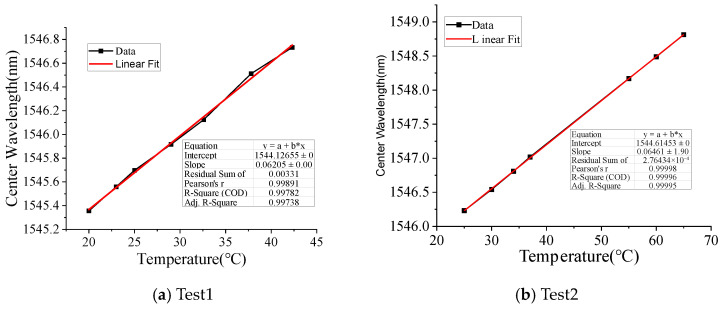
The temperature calibration result of LFBG.

**Figure 5 micromachines-16-00810-f005:**
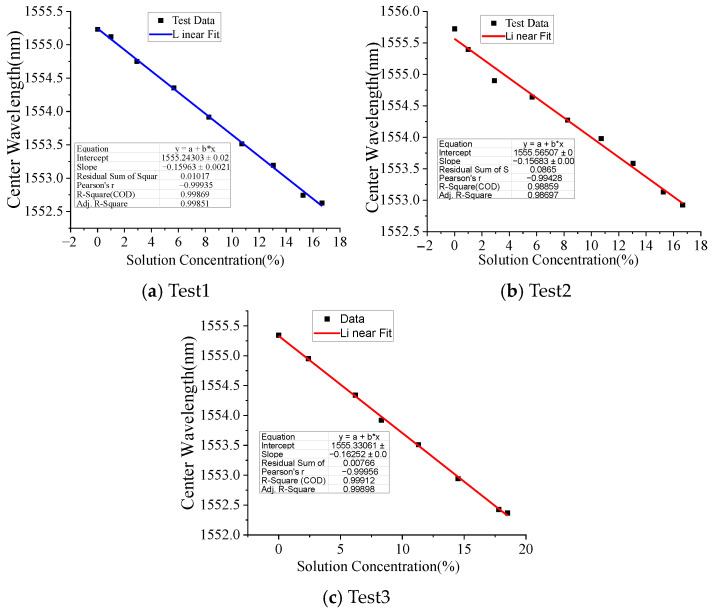
Linear correlation between LPFG central wavelength and NaCl solution concentration.

**Figure 6 micromachines-16-00810-f006:**
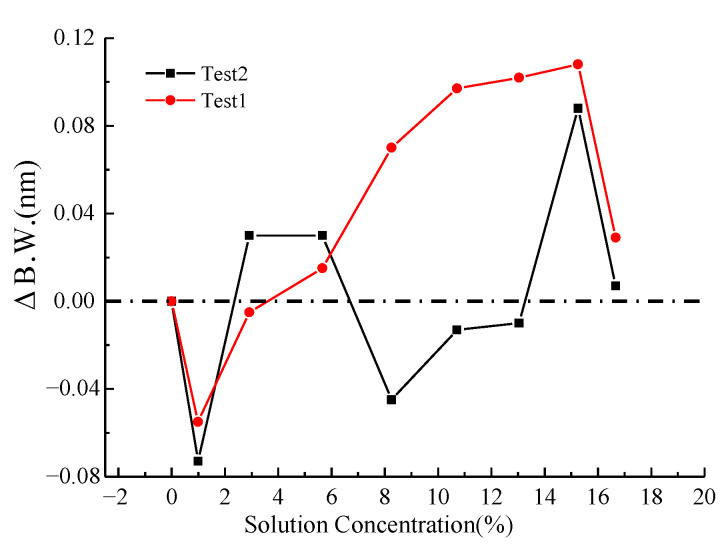
Three dB bandwidth variation with NaCl solution concentration.

**Figure 7 micromachines-16-00810-f007:**
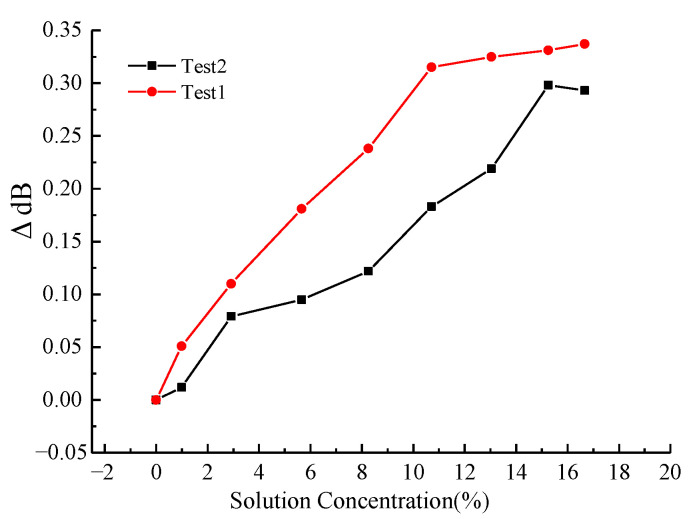
Peak loss variation with NaCl solution concentration.

**Figure 8 micromachines-16-00810-f008:**
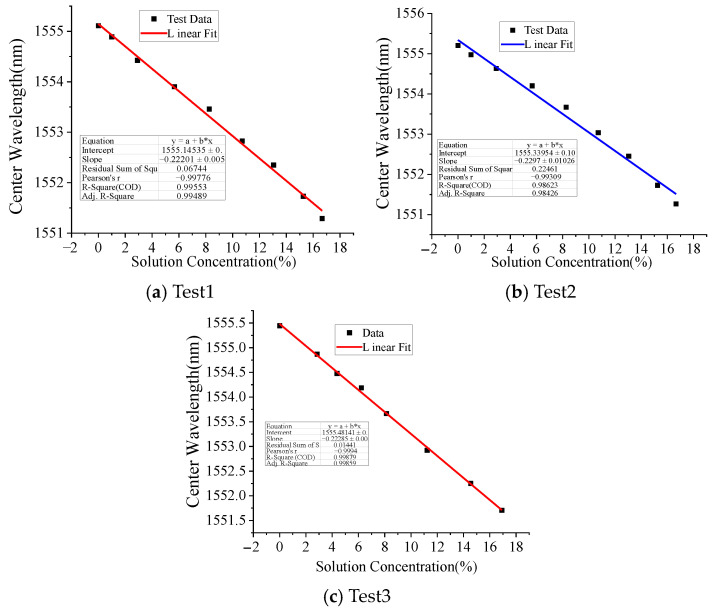
Linear correlation between LPFG central wavelength and CaCl_2_ solution concentration.

**Figure 9 micromachines-16-00810-f009:**
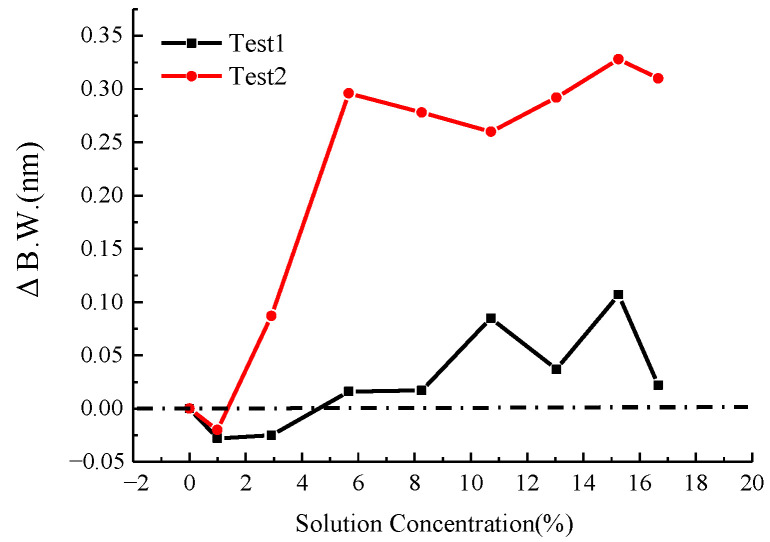
Three dB Bandwidth variation with CaCl_2_ solution concentration.

**Figure 10 micromachines-16-00810-f010:**
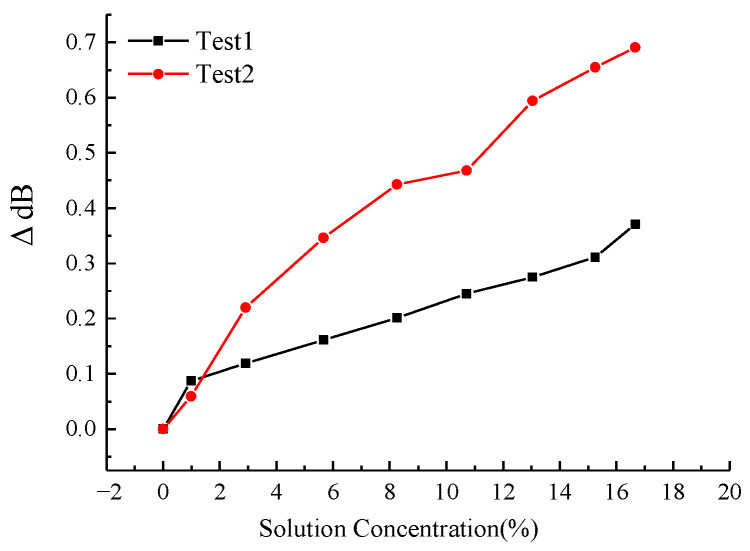
Peak loss variation with CaCl_2_ solution concentration.

**Figure 11 micromachines-16-00810-f011:**
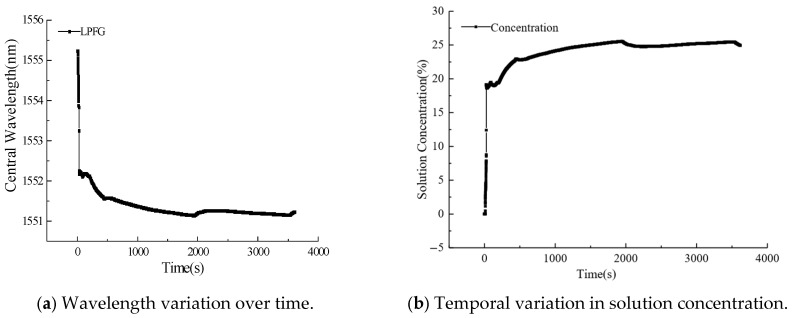
The results of real-time monitoring of sodium chloride solution concentration.

**Figure 12 micromachines-16-00810-f012:**
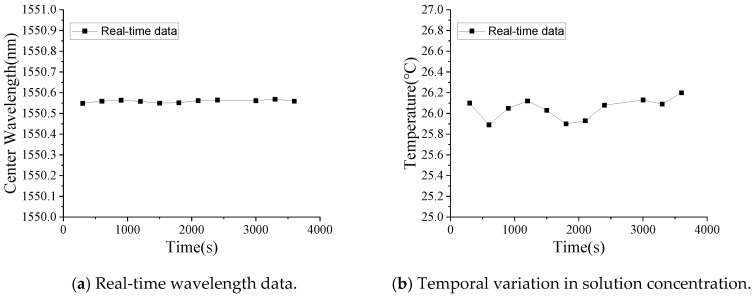
Real-time monitoring data for solution temperature.

**Table 1 micromachines-16-00810-t001:** Sensor sensing performance.

	NaCl Solution Concentration			CaCl_2_ Solution Concentration		
	Test1	Test2	Test3	Test1	Test2	Test3
Sensing coefficient (nm/%)	−0.160	−0.157	−0.163	−0.222	−0.230	−0.223
Residual sum of square	0.01017	0.0865	0.00766	0.06744	0.22461	0.01441
Pearson’s r	−0.99935	−0.99428	−0.99956	−0.99776	−0.99309	−0.9994
R-Square	0.99869	0.98859	0.99912	0.99553	0.98623	0.99879
Adj. R-Square	0.99851	0.98697	0.99898	0.99489	0.98426	0.99859

## Data Availability

The original contributions presented in the study are included in the article; further inquiries can be directed to the corresponding author.

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
