# Peer review of "Development of LPFG-Based Seawater Concentration Monitoring Sensors Packaged by BFRP"

_micromachines, 2025, doi:10.3390/mi16070810_

Round 1

Reviewer 1 Report

Comments and Suggestions for Authors

The authors aim to develop and validate a robust optical fiber sensor for monitoring salinity. The focus is the Long-Period Fiber Grating (LPFG) protection/encapsulation, as the authors claim the Basalt Fiber Reinforced Polymer (BFRP) shell as the novelty.

The quality and clarity must be improved. The manuscript is generally structured logically, but the language contains numerous grammatical errors and unsual phrasing ("gradually if the calcium chloride particles changed the concentration" , "center wavelength crest loss"). Figures are presented in low resolution and the spectrum is showed by picture of a screen.

Whereas the encapsulation is indeed interesting and novel, the authors present no evidence of its effectiveness. The introduction provides a general background but could better articulate the specific gap this sensor fills compared to other existing packaged fiber optic salinity sensors. The study should be redesigned to better express the probe design (encapsulation), compare with other types (or bare fiber), for example.

Scheme 2 shows no return of the optical fiber after the LPFG, a mirror was used? The authors should clarly state and also discuss the bend radius if the probe does not work in reflection.

I suggest the authors to redesign the study and focus on the encapsulation, for example: the conclusion that the BFRP package "ensures the stability of the sensor for long-term use" is an inference from material literature, not from long-term tests performed. Such tests would improve the reearch. The statstical rigor should also be improves. All calibration experiments should be repeated at least 3-5 times to allow for meaningful statistical analysis. The calibration graphs (Figures 5 and 8) must include error bars for each data point. The manuscript should discuss the standard deviation, reproducibility, and calculated limit of detection.

Author Response

Comments 1:

The authors aim to develop and validate a robust optical fiber sensor for monitoring salinity. The focus is the Long-Period Fiber Grating (LPFG) protection/encapsulation, as the authors claim the Basalt Fiber Reinforced Polymer (BFRP) shell as the novelty. The quality and clarity must be improved. The manuscript is generally structured logically, but the language contains numerous grammatical errors and unsual phrasing ("gradually if the calcium chloride particles changed the concentration" , "center wavelength crest loss"). Figures are presented in low resolution and the spectrum is showed by picture of a screen.

Response 1: Thanks, we carefully revised the article and improved the quality of the images

Comments 2Whereas the encapsulation is indeed interesting and novel, the authors present no evidence of its effectiveness. The introduction provides a general background but could better articulate the specific gap this sensor fills compared to other existing packaged fiber optic salinity sensors. The study should be redesigned to better express the probe design (encapsulation), compare with other types (or bare fiber), for example.

Response 2: We have added two references on fiber optic salinity sensors as following:

15]Qian, Yu, Yong Zhao, Qi-lu Wu, and Yang Yang. "Review of salinity measurement technology based on optical fiber sensor." Sensors and Actuators B: Chemical 260 (2018): 86-105.

16]Li, Gaochao, Yongjie Wang, Ancun Shi, Yuanhui Liu, and Fang Li. "Review of seawater fiber optic salinity sensors based on the refractive index detection principle." Sensors 23, no. 4 (2023): 2187.

The basalt fiber-reinforced polymer (BFRP) encapsulation demonstrates excellent durability in high-salinity and high-humidity corrosive conditions, eliminating the oxidation problems typical of traditional electrode-based sensors. In this study, the BFRP encapsulation technique effectively improves the sensor's measurement stability and service life longevity.

Comments 3Scheme 2 shows no return of the optical fiber after the LPFG, a mirror was used? The authors should clarly state and also discuss the bend radius if the probe does not work in reflection.

Response3: This was our mistake. We mistakenly placed a photo of the FBG salinity sensor developed by our research team in the article. We will replace it with a photo of the LPFG sensor in the revised article.

Figure.2

Comments 4 I suggest the authors to redesign the study and focus on the encapsulation, for example: the conclusion that the BFRP package "ensures the stability of the sensor for long-term use" is an inference from material literature, not from long-term tests performed. Such tests would improve the research. The statistical rigor should also be improves. All calibration experiments should be repeated at least 3-5 times to allow for meaningful statistical analysis. The calibration graphs (Figures 5 and 8) must include error bars for each data point. The manuscript should discuss the standard deviation, reproducibility, and calculated limit of detection.

Response4: Thanks. In the revised paper, we added an additional set of calibration tests for each test condition and analyzed the calibration results, which are listed in Table 1. Calibration test results show that the sensor has excellent sensing characteristics and linearity.

The additional calibration chart is shown in Figure 5(c) and Figure8(c)

Reviewer 2 Report

Comments and Suggestions for Authors

This manuscript presents a seawater salinity sensor based on Long-Period Fiber Gratings (LPFG), encapsulated using Basalt Fiber Reinforced Polymer (BFRP). The study includes sensor fabrication, calibration with NaCl and CaCl₂ solutions, and real-time concentration monitoring experiments. The topic is relevant and timely for the fields of fiber-optic sensing and marine environmental monitoring. The sensor design is likely, and the use of BFRP for encapsulation contributes to durability in corrosive environments. While the integration of LPFG and BFRP packaging for seawater concentration detection is technically interesting; however, the level of innovation appears limited, as similar LPFG-based sensors for corrosion, strain, and concentration monitoring have been extensively reported in previous literature. In addition, the manuscript has some issues in presentation and clarity. Below are detailed comments and suggestions.

  1. The authors should more clearly differentiate their work from existing studies (e.g., Ref. [15–17]). What’s the unique contribution here? Is it the encapsulation strategy, the real-time measurement, or the BFRP selection?
  2. While the use of NaCl and CaCl₂ solutions as simplified test media is understandable, have the authors considered or evaluated the actual refractive index range of real seawater? Since seawater contains a complex mixture of ions beyond just Na⁺ and Cl⁻, the optical properties may differ. It would be helpful to understand how representative the current test solutions are compared to real seawater conditions.
  3. The Section 1 (Introduction) lacks the discussion of the spectroscopic interference method (e.g., Scientific Reports, 15482 (2019)).
  4. The temperature compensation is only mentioned briefly via reference to the LPFG-FBG cascaded work, but is not clearly implemented in this study.
  5. Figure 2 lacks dimensions.
  6. Although Figures 5 and 8 show good linearity, the explanation of linearity, residuals, and measurement uncertainty is limited.
  7. In Figures 6, 7, 9, and 10 show poor linearity. Is there any practical use for those features? Can authors critically discuss this issue?
  8. The manuscript contains some awkward phrasing and unclear meaning (e.g., “high frequency CO₂ laser pulses,” “data from test1,” “the center wavelength crest loss”). Language polishing is necessary.
  9. Some figures are difficult to read, with insufficient resolution (e.g., Figs. 4, 5 and 8).
  10. The title (“R&D” of the LPFG-based…..) of Section 3 is not proper.
  11. In Section 4.3, the authors state that the ambient temperature is 26 °C. However, this does not guarantee that the solution temperature is also 26 °C. Additionally, the method for long-term temperature monitoring is not described.
Comments on the Quality of English Language

The manuscript contains some awkward phrasing and unclear meaning (e.g., “high frequency CO₂ laser pulses,” “data from test1,” “the center wavelength crest loss”). Language polishing is necessary.

Author Response

Comments 1:

The authors should more clearly differentiate their work from existing studies (e.g., Ref. [15–17]). What’s the unique contribution here? Is it the encapsulation strategy, the real-time measurement, or the BFRP selection?.

Response 1: Many thanks to the reviewer for his or her comments.

In the revised paper: Ref.18 and ref.19developed two type LPFG corrosion sensors coated with Ag and Fe-C layers respectively. As these protective layers undergo corrosion, they induce measurable changes in the surrounding refractive index at the LPFG surface. The disadvantage of this sensor is that it cannot be reused.

Ref.21 proposed a LPFG-FBG sensor for simultaneous measurement of temperature and humidity.

This study focuses on using BFRP to encapsulate LPFG sensors, improving the durability and stability of sensor measurements. The sensors themselves do not have a coating layer and are bidirectional reversible sensors.

Comments 2: While the use of NaCl and CaCl₂ solutions as simplified test media is understandable, have the authors considered or evaluated the actual refractive index range of real seawater? Since seawater contains a complex mixture of ions beyond just Na⁺ and Cl⁻, the optical properties may differ. It would be helpful to understand how representative the current test solutions are compared to real seawater conditions.

Response2: This is a very good question. In actual engineering applications, I first need to understand the monitoring environment and calibrate the sensor based on the solution in the monitoring area.

In this study, our main focus was on laboratory testing, and we did not consider or evaluate the refractive index range of actual seawater. Marine environments vary across different latitudes, so in practical applications, it is necessary to extract seawater from the monitoring environment for secondary calibration of the sensor in order to obtain more accurate monitoring data.

Comments 3:The Section 1 (Introduction) lacks the discussion of the spectroscopic interference method (e.g., Scientific Reports, 15482 (2019)).

Response3: Thanks, we have added this paper in the reference lists, and we also added some content about this method in the introduction section.

“Among these methods, the interference method is one of the most sensitive methods for measuring the refractive index of seawater, which has broad-range adaptability (Refractive index measurement range of 1.33-1.38) ;”

Ref. 1: Uchida, H., Kayukawa, Y. & Maeda, Y. Ultra high-resolution seawater density sensor based on a refractive index measurement using the spectroscopic interference method. Sci Rep 9, 15482 (2019).

Comments 4:The temperature compensation is only mentioned briefly via reference to the LPFG-FBG cascaded work, but is not clearly implemented in this study.

Response4: Thanks. In current sensing and monitoring applications, dual-parameter compensation has emerged as the predominant approach for temperature compensation. This methodology typically employs two distinct sensing mechanisms: one dedicated to temperature measurement and another targeting the specific parameter of interest. Through differential signal processing, the temperature-induced variations can be effectively decoupled from the target parameter measurements, thereby achieving accurate temperature compensation. The authors have also proposed dual-sensing monitoring methods based on BOTDR-FBG and BOTDR-DTS. (ref.18 Liu, B.; He, J.P.; Zhang, L.; Tang, J.L.; et. al. Pipeline safety monitoring technology based on FBG-ROTDR joint system and its case study of urban drainage pipeline monitoring, Optical Fiber Technology, 73(2022)103044.)

In this paper, all experiments were conducted indoors, where the indoor environment remained largely unchanged. Therefore, environmental temperature compensation was not considered in the experiments.

Comments 5: Figure 2 lacks dimensions.

Response 5: Thanks. the diameter of the sensor is 5mm.

Comments 6: Although Figures 5 and 8 show good linearity, the explanation of linearity, residuals, and measurement uncertainty is limited.

Response 6 : Thanks. we have added Table.1 to list the linearity, residuals, and measurement uncertainty.

Comments 7: In Figures 6, 7, 9, and 10 show poor linearity. Is there any practical use for those features? Can authors critically discuss this issue?

Response7: Thanks. The results of the above calibration tests show that although the center wavelength, spectral 3dB bandwidth, and spectral peak loss are all related to changes in solution concentration, only the center wavelength is linearly related to solution concentration. In the revised paper, we have added this content.

Comments 8: The manuscript contains some awkward phrasing and unclear meaning (e.g., “high frequency CO₂ laser pulses,” “data from test1,” “the center wavelength crest loss”). Language polishing is necessary.

Response8: Thanks. We have carefully revised the article for spelling, grammar, and wording.

Comments 9: Some figures are difficult to read, with insufficient resolution (e.g., Figs. 4, 5 and 8).

Response 9: Thanks. We have improved the figures in the revised paper.

Comments10:      The title (“R&D” of the LPFG-based…..) of Section 3 is not proper.

Response10: We replaced the title with “Sensor Design”

Comments 11:      In Section 4.3, the authors state that the ambient temperature is 26 °C. However, this does not guarantee that the solution temperature is also 26 °C. Additionally, the method for long-term temperature monitoring is not described.

Response 11: Before starting the experiment, 500 ml of water was left to stand indoors for one hour to ensure that the water temperature was consistent with the room temperature.

We monitored the temperature of the solution in real time using an LPFG temperature sensor placed in a capillary tube. Since the data remained largely unchanged, we did not include this data in the first draft of the manuscript. In the second draft, we added this experimental data.

“Figure 12 shows the real-time monitoring data for solution temperature measured by LPFG sensor encapsulated in a capillary tube, and the sampling time is every 300 seconds. It can be seen that the solution temperature is basically maintained at 26°C.”

Round 2

Reviewer 1 Report

Comments and Suggestions for Authors

Thank you for addressing the suggestions.

Reviewer 2 Report

Comments and Suggestions for Authors

All previous concerns have been addressed. I have no further comments.